

# Interdisciplinary, interinstitutional and international collaboration of family medicine researchers in Taiwan

Yi-Hsuan Lin[1,2], Yen-Han Tseng[3,4], Hsiao-Ting Chang[2,5], Ming-Hwai Lin[2,5], Yen-Chiang Tseng[4,6], Tzeng-Ji Chen[2,5,7] and Shinn-Jang Hwang[2,5]

[1] Division of Family Medicine, Taipei Hospital, Ministry of Health and Welfare, New Taipei City, Taiwan
[2] Department of Family Medicine, School of Medicine, National Yang-Ming University, Taipei, Taiwan
[3] Chest Department, Taipei Veterans General Hospital, Taipei, Taiwan
[4] School of Medicine, National Yang-Ming University, Taipei, Taiwan
[5] Department of Family Medicine, Taipei Veterans General Hospital, Taipei, Taiwan
[6] Department of Surgery, Kaohsiung Veterans General Hospital, Pingtung Branch, Pingtung, Taiwan
[7] Institute of Hospital and Health Care Administration, School of Medicine, National Yang-Ming University, Taipei, Taiwan

Corresponding author
Tzeng-Ji Chen,
tjchen@vghtpe.gov.tw

## ABSTRACT

The family medicine researches flourished worldwide in the past decade. However, the collaborative patterns of family medicine publications had not been reported. Our study analyzed the collaborative activity of family medicine researchers in Taiwan. We focused on the types of collaboration among disciplines, institutions and countries. We searched "family medicine" AND "Taiwan" in address field from Web of Science and documented the disciplines, institutions and countries of all authors. We analyzed the collaborative patterns of family medicine researchers in Taiwan from 2010 to 2014. The journal's impact factor of each article in the same publication year was also retrieved. Among 1,217 articles from 2010 to 2014, interdisciplinary collaboration existed in 1,185 (97.3%) articles, interinstitutional in 1,012 (83.2%) and international in 142 (11.7%). Public health was the most common collaborative discipline. All international researches were also interdisciplinary and interinstitutional. The United States (75 articles), the United Kingdom (21) and the People's Republic of China (20) were the top three countries with which family medicine researchers in Taiwan had collaborated. We found a high degree of interdisciplinary and interinstitutional collaboration of family medicine researches in Taiwan. However, the collaboration of family medicine researchers in Taiwan with family medicine colleagues of other domestic or foreign institutions was insufficient. The future direction of family medicine studies could focus on the promotion of communication among family medicine researchers.

## INTRODUCTION

Family physicians are the main providers of primary health care. Although in most countries family physicians belong to the largest group of medical specialties, family medicine research did not receive due attention (*Bolon & Phillips, 2010*). Academic performance assessment by SCI (Scientific Citation Index) and IF (impact factor) also affected the development of family medicine research (*Lin et al., 2006*). To publish more papers in journals of higher IFs, in most cases in other specialties or disciplines, many family medicine researchers tried to establish a collaborative research network (*Parchman, Katerndahl & Larme, 2003*). In the past decade, publications from family medicine researchers did increase rapidly (*Abdulmajeed, Ismail & Nour-Eldein, 2014*; *Lin et al., 2014*). However, their patterns of collaboration have been rarely analyzed.

The aim of the current study was to investigate the collaborative patterns of family medicine researchers in Taiwan. Family medicine has become an official specialty in Taiwan since 1988 (*Department of Health, Executive Yuan, 1988*). Up to July 2015, there were 5,075 family medicine specialists in Taiwan (*Ministry of Health and Welfare, 2015*). According to the statistics in December 2013, the residency training programs for family medicine were carried out in 83 hospitals with 493 attending physicians (*Taiwan Association of Family Medicine, 2015*). We would analyze the collaborative activity of family medicine researchers in the recent 5 years. Instead of interpersonal aspect, the collaboration would be examined among disciplines, institutions and nations. The descriptive analyses might carry implications for the future of family medicine research.

## MATERIALS AND METHODS

### Subjects and data collection

We downloaded the records from Web of Science® with Science Citation Index Expanded (SCI) and Social Science Citation Index (SSCI) databases in April 2015. The search criteria were "family medicine" AND "Taiwan" in address. We analyzed only articles and excluded other publication forms such as abstracts, letters, notes, news, editorials, meeting summaries and reviews. As long as a family medicine physician was listed in co-author in an article, we enrolled this article and analyzed its collaborative patterns. Our search results might include papers in which family medicine researchers in other countries collaborated with non-family medicine researchers in Taiwan. These papers were manually screened out.

### Study design

In each article, we documented the disciplines, institutions and countries of all authors in addition to publication year, journal's name and IF. We also identified the articles with family medicine researchers as the first or corresponding authors. The journal's IF of each article in the same publication year was retrieved from Web of Science. Because the Journal Citation Reports 2014 was not available at the time of study execution, the IFs of articles in 2014 were substituted with the IFs in 2013.

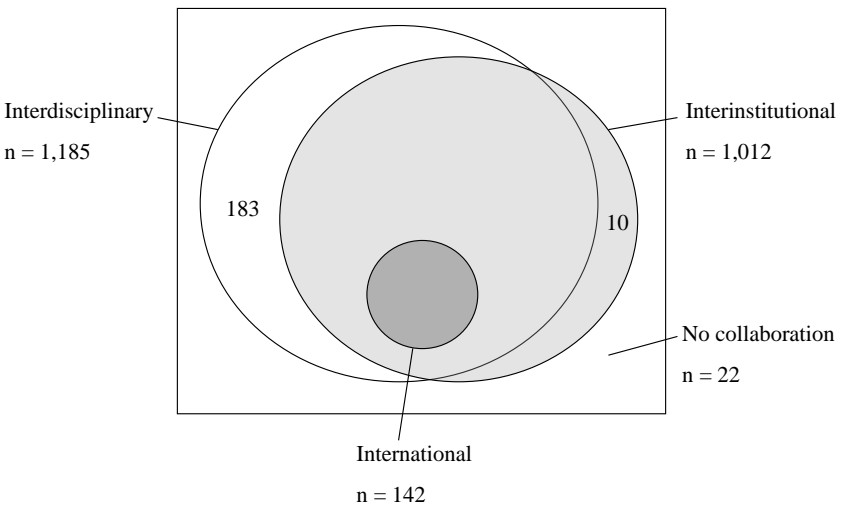

**Figure 1 Interdisciplinary, interinstitutional and international collaboration of family medicine researchers in Taiwan from 2010 to 2014.**

In addition to the analysis of publication trend of family medicine researchers in Taiwan from 1993 to 2014, we calculated the interdisciplinary, interinstitutional and international collaborative patterns of family medicine researchers in Taiwan from 2010 to 2014. The collaborations were also stratified by institution of family medicine researchers.

## Statistical analyses

Descriptive statistics were presented. All calculations were performed with Microsoft Excel 2013.

## RESULTS

From 1993 to 2014, family medicine researchers in Taiwan had authored 2,107 articles. The annual publication counts in the early 5 years were less than 10 articles per year, increased to 101 articles in 2007 and continuously grew to be more than 200 articles in 2011.

Among 1,217 articles from 2010 to 2014, family medicine researchers in Taiwan acted as the first authors in 443 (36.4%) articles, corresponding authors in 347 (28.5%) articles, and both the first and corresponding authors in 198 articles (16.3%). The overwhelming majority of publications were results of collaboration and only 22 articles came from a single family medicine department. Interdisciplinary collaboration existed in 1,185 (97.3%) articles, interinstitutional in 1,012 (83.2%) and international in 142 (11.7%). In 1,002 (82.3%) articles there were both interdisciplinary and interinstitutional collaborations. All 142 articles with international collaboration had also interdisciplinary and interinstitutional collaborations (Fig. 1).

From 2010 to 2014, family medicine researchers from 71 institutions in Taiwan had published articles indexed in the Web of Science. All articles had on average an IF of 2.9 (SD 2.2) and only 112 (9.2%) articles had an IF $\geq$5. Table 1 showed the paper counts with mean journal IF in the top 20 institutions. Taipei Veterans General Hospital was most productive with 215 (17.7%) articles, followed by Chung Shan Medical University Hospital with 140

**Table 1** The interdisciplinary, interinstitutional and international collaboration and the mean IF of the top 20 institutions of family medicine researchers in Taiwan from 2010 to 2014.

| Institutions of family medicine researchers | No. of articles | IF (Mean ± SD) | No. of articles with IF >= 5 (%, $n = 112$) | Interdisciplinary No. of articles (%) | Interinstitutional No. of articles (%) | International No. of articles (%) |
|---|---|---|---|---|---|---|
| Taipei Veterans General Hospital | 215 | 3.45 ± 2.40 | 39 (34.8) | 214 (99.5) | 169 (78.6) | 24 (11.2) |
| Chung Shan Medical University Hospital | 140 | 2.84 ± 3.22 | 8 (7.1) | 140 (100) | 104 (74.3) | 5 (3.6) |
| China Medical University Hospital | 136 | 2.95 ± 2.11 | 14 (12.5) | 135 (99.3) | 122 (89.7) | 21 (15.4) |
| National Taiwan University Hospital | 111 | 2.87 ± 2.22 | 8 (7.1) | 108 (97.3) | 94 (84.7) | 18 (16.2) |
| National Cheng Kung University Hospital | 67 | 3.34 ± 2.25 | 10 (8.9) | 54 (80.6) | 36 (53.7) | 8 (11.9) |
| Kaohsiung Medical University Hospital | 58 | 2.84 ± 1.43 | 5 (4.5) | 58 (100) | 46 (79.3) | 10 (17.2) |
| Taipei Medical University Hospital | 49 | 3.48 ± 3.29 | 5 (4.5) | 49 (100) | 46 (93.9) | 4 (8.2) |
| Chang Gung Memorial Hospital | 45 | 2.76 ± 1.70 | 6 (5.4) | 44 (97.8) | 38 (84.4) | 14 (31.1) |
| Buddhist Tzu Chi General Hospital | 35 | 2.37 ± 1.14 | 0 | 33 (94.3) | 26 (74.3) | 7 (20.0) |
| Taipei City Hospital | 32 | 2.84 ± 2.36 | 2 (1.8) | 32 (100) | 32 (100) | 7 (21.9) |
| Taichung Veterans General Hospital | 28 | 2.48 ± 1.41 | 1 (0.9) | 28 (100) | 26 (92.9) | 2 (7.1) |
| Tri-Service General Hospital | 27 | 2.81 ± 1.55 | 2 (1.8) | 26 (96.3) | 20 (74.1) | 3 (11.1) |
| Chi Mei Medical Center | 26 | 2.48 ± 1.27 | 1 (0.9) | 26 (100) | 25 (96.2) | 4 (15.4) |
| Cardinal Tien Hospital | 25 | 2.27 ± 1.35 | 1 (0.9) | 25 (100) | 22 (88.0) | 1 (4.0) |
| Armed Forces General Hospital | 19 | 1.97 ± 0.94 | 0 | 17 (89.5) | 15 (78.9) | 0 |
| Mackay Memoral Hospital | 17 | 1.58 ± 1.46 | 0 | 16 (94.1) | 13 (76.5) | 0 |
| Kaohsiung Veterans General Hospital | 16 | 1.56 ± 1.09 | 0 | 16 (100) | 16 (100) | 2 (12.5) |
| Cathay General Hospital | 12 | 3.95 ± 2.25 | 2 (1.8) | 12 (100) | 11 (91.7) | 0 |
| Ministry of Health and Welfare, Taichung Hospital | 12 | 1.72 ± 1.21 | 0 | 12 (100) | 12 (100) | 1 (8.3) |
| Chia Yi Christian Hospital | 11 | 3.04 ± 1.63 | 1 (0.9) | 11 (100) | 11 (100) | 0 |

(11.5%) and China Medical University Hospital with 136 (11.2%). Taipei Veterans General Hospital had also the largest number of papers with IF ≥5 (39 articles).

Public health is the discipline with which family medicine researchers in Taiwan mostly collaborated (493 articles, 40.5%), almost two and a half times as much as that with internal medicine (16.4%). Most institutions with family medicine researchers in Taiwan had a high percentage of interdisciplinary and interinstitutional collaborations, with the exception of National Cheng Kung University Hospital (interinstitutional collaboration in only 53.7% of articles) (Table 1). The Chang Gung Memorial Hospital had the highest percentage (31.1%) of international collaboration.

Table 2 showed the geographic distribution of international collaboration of family medicine researchers in Taiwan. Most research partners existed in North America (84 articles), followed by Europe (43) and Asia (26). The United States (75 articles), the United Kingdom (21) and the People's Republic of China (20) were the top three countries with which family medicine researchers in Taiwan had collaborated.

**Table 2** Distribution of international collaboration of family medicine researchers in Taiwan, stratified by continent and country.

| Continent | No. of articles | Country | No. of articles |
|---|---|---|---|
| North America | 84 | United States | 75 |
| | | Canada | 11 |
| Europe | 43 | United Kingdom | 21 |
| | | Germany | 19 |
| | | Netherlands | 6 |
| | | Belgium | 3 |
| | | France | 3 |
| | | Poland | 2 |
| | | Italy | 2 |
| | | Czech Republic | 1 |
| | | Hungary | 1 |
| | | Sweden | 1 |
| | | Spain | 1 |
| | | Switzerland | 1 |
| | | Bulgaria | 1 |
| | | Norway | 1 |
| Asia | 26 | Peoples R China | 20 |
| | | Japan | 7 |
| | | Malaysia | 4 |
| | | South Korea | 3 |
| | | Indonesia | 3 |
| | | Singapore | 3 |
| | | India | 2 |
| | | Philippines | 2 |
| | | Thailand | 2 |
| | | Israel | 1 |
| Oceania | 9 | Australia | 8 |
| | | New Zealand | 1 |
| Latin America | 4 | Mexico | 2 |
| | | Nicaragua | 1 |
| | | Brazil | 1 |

## DISCUSSION

Our study was a descriptive analysis to investigate the collaboration of family medicine research in Taiwan. The collaborative activity was based on authorship (*Katz & Martin, 1997*) and the collaborative types were divided into interdisciplinary, interinstitutional and international. We found most publications by family medicine researchers in Taiwan had interdisciplinary and interinstitutional collaboration. However, the collaboration of family medicine researchers in Taiwan with family medicine colleagues of other domestic or foreign institutions was relatively low.

Our results showed almost all publications from family medicine researchers in Taiwan had collaborative relationship with other disciplines and were published in subject categories other than "primary health care." This phenomenon might be related to the influence of impact factors and publication volumes which are relatively low in family medicine journals (*Lin et al., 2006*). The family medicine research always faces great challenges. In 2003, the World Organization of Family Doctors (Wonca) conference recommended to build the research capacity of this discipline (*Van Weel & Rosser, 2004*). *Voorhees et al. (2013)* reported that only 4.9% of the 28,505 board-certified family physicians in the United States had spent time for research. Lack of time, funding support and resources blocked the development of family medicine researchers and the collaborative relationship with other disciplines might help to solve the problem (*Beasley, 2011*). Interdisciplinary collaboration did offer better accessibility of resources (*Beasley, 2011*). In Sweden, the National Research School in General Practice provided a creative environment for multidisciplinary researches (*Horton, 2014*). On the other side, the research groups among family medicine researchers also flourished worldwide, e.g., the South Asia Primary Care Research Network, the North America Primary Care Research Group, and the European General Practice Research Network (*Kidd et al., 2014*). However, our study revealed that the research network among family medicine researchers in Taiwan didn't develop well and the thriving research output was attributed to the interdisciplinary activities.

Few studies had analyzed the collaborative disciplines of family medicine researchers. Our study displayed that public health was the most common collaborative discipline of family medicine researchers in Taiwan. Family medicine and public health are alleged to be natural allies (*Campos-Outcalt, 2004*). Public health promotes the health of public, including epidemiology and preventive medicine. The goal of public health is also pursued by family physicians in the daily practice, e.g., health education, disease screening, immunization and smoking cessation (*Campos-Outcalt, 2004*). It can be understood that the two disciplines collaborate more effectively (*Kempe et al., 2014*).

A report showed that 15.6% articles in the SCI database were internationally co-authored (*Wagner & Leydesdorff, 2005*). In Taiwan, the share of clinical medicine researches with international cooperation was 13.6% during 1990–2004 (*Chen et al., 2007*). In our results, the average international collaborative rate was 11.7%. International research collaboration could increase the scientific popularity and visibility (*Persson, Glanzel & Danell, 2004*; *Chinchilla-Rodriguez et al., 2010*). Articles with international collaboration were usually published in journals with higher impact factors or more frequently cited (*Chen et al., 2007*; *Persson, Glanzel & Danell, 2004*). In addition, collaboration among researchers with diverse scientific backgrounds might provide more resources, enhance sharing and transfer of knowledge, skills and techniques, and bring more stimulation and creativity (*Katz & Martin, 1997*). On the other hand, researchers might lose the true meaning of research if the purpose of collaboration was just to get access to high IF journals. It would be thus hard to maintain the specificity of the discipline and country.

Special attention should be paid to Fig. 1 which showed that all international family medicine researches in Taiwan belonged to interdisciplinary articles. That is, family

medicine researchers in Taiwan did not join the research network of family medicine in other countries or international organizations. Because the research materials of family medicine usually come from daily primary care (*De Lusignan et al., 2011*), the differences of healthcare systems might hinder the data integration in international research. Even though Wonca has organized several working panels to actively promote international collaboration, family medicine researchers in Taiwan didn't seem to have taken advantage of these opportunities or achieved successful outcomes.

The international collaboration of family medicine researchers in Taiwan was not geographically limited. In the past the collaborative research activities could be affected by the geographic distance and in the internet era this limitation was effaced (*Katz & Martin, 1997*). Situated in Far East, Taiwan had closer collaboration with North America and Europe than with Asian countries. Even though Taiwan and the People's Republic of China share the same cultural origin, the collaboration of both sides was far from satisfactory. The undesirable collaboration also applied to Japan, which had colonized Taiwan for fifty years. Paraje and colleagues (*2009*) had reported similar results. In the Western Pacific region, the intra-regional collaboration of health research was low. Countries with large research output (e.g., Japan, China) collaborated more with high-income countries from other regions (*Paraje, Sadana & Salmela, 2009*).

Our study revealed the insufficient collaboration of family medicine researchers in Taiwan with family medicine colleagues of other institutions, either domestic or foreign. It also existed in other specialties and countries (*Paraje, Sadana & Salmela, 2009*). The Wonca conference in 2003 developed some recommendations to solve this problem. One major step was to organize a platform for communication among family medicine researchers, either provided by national medical associations, university departments of family medicine, or family medicine research institutes. The collaborative platform could be extended from local to national and regional levels (*Van Weel & Rosser, 2004*).

There were some limitations in our study. Our search criteria were "family medicine" AND "Taiwan" in the address field of Web of Science. If a department with family physicians carried other names, e.g., community medicine, its publications might be missed. The possibility should be low because all family medicine departments of larger academic medical centers in Taiwan were enrolled in our study. Furthermore, we analyzed the collaborative activity according to the affiliations of authors. Authors with multiple affiliations might result in overestimation of collaboration. The possibility should be low because multiple authorship prevailed so that interpersonal collaboration in this situation would usually imply interdisciplinary or interinstitutional collaboration. In spite of these limitations, our article was the first study to report the collaborative patterns of family medicine researchers from interdisciplinary, interinstitutional and international levels. Our finding was helpful to the discussions of current scientific research collaborative status.

## CONCLUSIONS

Previous study (*Lin et al., 2006*) found that the academic performance assessment caused family medicine researchers in Taiwan to publish more papers in non-family medicine,

foreign, English-language, and SCI-indexed journals. Our current study revealed that family medicine researchers in Taiwan rapidly expanded their publications through a high degree of interdisciplinary and interinstitutional collaboration. However, the collaboration of family medicine researchers in Taiwan with family medicine colleagues of other institutions, either domestic or foreign, lagged far behind. The organization of a platform to enhance the communication among family medicine researchers might be a solution.

## ACKNOWLEDGEMENT

The authors would like to thank Ching Wen Liang for computer graphics.

### Funding

The authors received no funding for this work.

### Competing Interests

The authors declare there are no competing interests.

### Author Contributions

- Yi-Hsuan Lin conceived and designed the experiments, performed the experiments, analyzed the data, wrote the paper, prepared figures and/or tables.
- Yen-Han Tseng conceived and designed the experiments, performed the experiments, analyzed the data, prepared figures and/or tables.
- Hsiao-Ting Chang, Ming-Hwai Lin, Yen-Chiang Tseng and Shinn-Jang Hwang contributed reagents/materials/analysis tools.
- Tzeng-Ji Chen conceived and designed the experiments, analyzed the data, wrote the paper, reviewed drafts of the paper.

### Supplemental Information

Supplemental information for this article can be found online at http://dx.doi.org/10.7717/peerj.1321#supplemental-information.

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
