# Peer review of "Interdisciplinary, interinstitutional and international collaboration of family medicine researchers in Taiwan"

_PeerJ, doi:10.7717/peerj.1321_

## Round 0.1 · original submission · Major Revisions

Please pay special attention to the comments of Reviewer #3.

Reviewer 1 ·

Basic reporting

The article addresses an important area for the family medicine research arena. The importance of collaboration in family medicine is important. However, the authors seem to lack to be critical to the fact that almost all the high-level publications from Taiwan have been made in collaboration with other disciplines. This in my view, is perhaps a sign for concern, because it may reflect the fact that family medicine researchers do not make research in the area of family medicine, but rather prefer to serve as collaborators in other areas of medicine. This may not be true, but the authors should at least address this issue.

Experimental design

The design is appropriate: a bibliometric reserach has been done.

Validity of the findings

The data are valid, but I would welcome more criticism regarding its validity. The issue of collaboration with other disciplines has been raised already. I would like the authors to give more reasons why do they feel that collaboration is necessary. Is the purpose of collaborating just to have better access to high IF journals? If it is, we are missing the point here.
On the other hand, collaboration with family medicine researchers outside Taiwan is important. But the authors should discuss the benefits (and potential pitfalls) regarding this collaboration. In my view, it is essential that the country and the discipline maintain the specificity of the country and the discipline and not get lost in internationalisation for the sake of having better access to high level publications.

·

Basic reporting

Comprehensive descriptive analysis on collaboration of family medicine researchers in Taiwan.

Experimental design

No comments

Validity of the findings

No comments

Additional comments

Within the section "Discussion", at the bottom of the page 9, the sentence starting with "The goal of public health is...", need to be supplemented by adding the words "health education" after the abbreviation "e.g.".

Reviewer 3 ·

Basic reporting

• In the “Abstract” section, the time range for the study is not mentioned.
• In the “Introduction” section, a short description of how Family Medicine is organized in Taiwan, in order to allow the readers outside the country to better understand the article.
• In the “Introduction” section, the authors should clarify the aim of their study.
• The authors are also advised to reduce number of Figures and Tables (data has already been mentioned in text). Leave as they are the Fig. 2 and Tbl 3 and 4, which contain the most relevant results of this study

Experimental design

• The authors didn't clearly define the research question.
• The methods should be clarified more precisely. If the family physician was the second, third or last author of an article, then why wouldn’t that article be classified as a family medicine research?

Validity of the findings

• Who is this paper meant to address, and what does it propose that [they] do regarding the paper’s findings?

Additional comments

The issue that you discussed here is relevant and interesting.
However, in order to improve the user value of your article, you need to enhance some parts, as well as alter a few. The article should be of interest to all readers worldwide, not just those within your own country (or even region), since the problem of insufficient scientific collaboration is globally relevant, especially in this area of expertise.

The description of the family medicine organization is lacking – for example, how many doctors are employed in the field of family medicine, how many of them actually have a relevant specialization, how many of them have an academic rank and which rank is it? Is family medicine taught as an independent class at medical schools?

It would improve the article is unnecessary details and listings were removed, and more attention was paid to the subject of collaboration (both domestic and international) among family medicine researchers.
How is this situation in other Asian countries that are similar to Taiwan in their location, size and/or population?

Are there any suggestions for improving this communication and collaboration? Who should do that and how? These should all be elaborated on.

Regarding the article conclusion: „However, domestic and international collaboration among family medicine researchers lagged far behind. The underlying causes deserve further investigations.”

This conclusion is too vague and general. Please rewrite it in the light of the previous considerations in this review.

---

## Round 0.2 · accepted · Accept

Please note the comment of the reviewer about proper spelling of the World organization of family doctors.

Reviewer 1 ·

Basic reporting

No comments

Experimental design

No comments

Validity of the findings

No comments

Additional comments

The paper is adequately corrected.
Just a minor comment: the abbreviation for the World organisation of family doctors is Wonca, not WONCA.

·

Basic reporting

No comments

Experimental design

No comments

Validity of the findings

No comments

Additional comments

No comments

Reviewer 3 ·

Basic reporting

No Comments

Experimental design

No Comments

Validity of the findings

No Comments